# The Role of N^6^-Methyladenosine (m^6^A) Methylation Modifications in Hematological Malignancies

**DOI:** 10.3390/cancers14020332

**Published:** 2022-01-11

**Authors:** Yan Zhao, Hongling Peng

**Affiliations:** 1Hunan Province Key Laboratory of Basic and Applied Hematology, Department of Hematology, The Second Xiangya Hospital, Central South University, Changsha 410011, China; Yan.Zhao@csu.edu.cn; 2Institute of Hematology, Central South University, Changsha 410011, China; 3Hunan Key Laboratory of Tumor Models and Individualized Medicine, Changsha 410011, China

**Keywords:** epigenetics, N^6^-methyladenosine, normal hematopoiesis, leukemia, multiple myeloma, myelodysplastic syndrome

## Abstract

**Simple Summary:**

Recently, despite the common application of various novel therapies (e.g., immunotherapy and stem cell transplantation) in hematologic tumors, hematologic malignancies remain suboptimal and have a worse prognosis due to the lack of donors and their high heterogeneity. Among them, epigenetic alterations (e.g., the abnormal modification of m^6^A) are essential to facilitate the progression of tumors and drug resistance. Our purpose in this study is to pinpoint the molecular targets of pathogenesis, as well as to analyze the oncogenic characteristics of m^6^A modifications. In this article, we, therefore, elaborate on the mechanisms of m^6^A modification and its role in normal hematopoietic regulation and malignant tumorigenesis, thus contributing to the refinement of molecularly targeted therapies.

**Abstract:**

Epigenetics is identified as the study of heritable modifications in gene expression and regulation that do not involve DNA sequence alterations, such as DNA methylation, histone modifications, etc. Importantly, N^6^-methyladenosine (m^6^A) methylation modification is one of the most common epigenetic modifications of eukaryotic messenger RNA (mRNA), which plays a key role in various cellular processes. It can not only mediate various RNA metabolic processes such as RNA splicing, translation, and decay under the catalytic regulation of related enzymes but can also affect the normal development of bone marrow hematopoiesis by regulating the self-renewal, proliferation, and differentiation of pluripotent stem cells in the hematopoietic microenvironment of bone marrow. In recent years, numerous studies have demonstrated that m^6^A methylation modifications play an important role in the development and progression of hematologic malignancies (e.g., leukemia, lymphoma, myelodysplastic syndromes [MDS], multiple myeloma [MM], etc.). Targeting the inhibition of m^6^A-associated factors can contribute to increased susceptibility of patients with hematologic malignancies to therapeutic agents. Therefore, this review elaborates on the biological characteristics and normal hematopoietic regulatory functions of m^6^A methylation modifications and their role in the pathogenesis of hematologic malignancies.

## 1. Introduction

Hematologic malignancies are a wide group of malignant clonal disorders of hematopoietic cells. Importantly, they typically manifest as uncontrolled proliferation and differentiation disorders of the hematopoietic cells, with high heterogeneity and poor prognosis, and so far, there is the lack of an effective cure. Currently, emerging evidence has revealed that the incidence of hematological malignancies ranks 6th in the total incidence of cancer in the world and 1st in the mortality rate of malignant tumors in adolescents [1]. Hematologic tumors are the second most common cause of cancer death in the United States. The occurrence of hematologic tumors is a complex multi-step process, and the malignancies mainly include lymphoma, leukemia, and myeloma, among other hematologic malignancies. In recent decades, studies [2,3] have shown that the pathogenesis of such diseases is associated with a variety of factors, including genomic abnormalities, epigenetic alterations, abnormal regulation of the bone marrow hematopoietic microenvironment, and disorders of the immune system.

In addition to the increasing number of patients suffering from this malignancy each year [1], inappropriate treatment modalities also lead to an annual increase in the mortality rate of this disease, which has inspired doctors to look for better treatment strategies. Notably, the progress of gene editing technologies and their application in the field of hematological tumors over the past decades have led to significant developments in the diagnosis and treatment of these malignancies [2]. Meanwhile, these findings shed light on the different mechanisms involved in the pathogenesis of hematologic malignancies [3]. Nevertheless, while most previous studies have focused on the study of genetic mechanisms of hematologic malignancies [4], recently, more attention has been paid to the field of epigenetics, which involves many closely related mechanisms regulating changes in gene expression levels without involving alterations in the DNA sequence [5].

In recent decades, epigenetics has been identified as a novel concept that corresponds to genetics and encompasses many types of well-documented epigenetic modifications. In plain words, epigenetic modification is a form of gene expression regulation that affects gene transcription and translation without changes in the nucleotide sequence and can regulate and thus affect gene expression at the level of DNA and chromatin structural modifications, RNA stability, and transcriptional activity, including DNA methylation modifications, histone covalent modifications, chromatin remodeling, non-coding RNA regulation, RNA modifications, etc. [6,7,8]. Further, epitranscriptomics is one of the newly emerging hot fields, which mainly focuses on the effects of chemical modifications carried by RNAs and their correlated regulators on gene expression. To date, it has been reported that more than one hundred chemical modifications (e.g., m^6^A) have been identified on RNAs, which perform extremely significant biological functions in living organisms via their involvement in mediating epigenetic regulation. More importantly, in recent years, epigenetic modifications have played a highly important role in the development and progression of hematologic malignancies (Figure 1).

At present, epitranscriptomics, represented by N^6^-methyladenosine (m^6^A) modifications, has become a hot topic of research due to the current study of second-generation sequencing, i.e., targeting their epigenetic alterations at the transcriptome level [9,10,11,12]. In essence, m^6^A methylation modifications can mediate the post-transcriptional regulation of gene expression without altering base sequences. In addition, RNA m^6^A modification is reversible and dynamically modulated by m^6^A modifiers (e.g., writers, erasers, and readers), which have currently been proven to play an important role in regulating mRNA decay, stability, variable splicing, translation efficiency, and localization [11,13,14]. Moreover, m^6^A sites are also present in long non-coding RNAs as well as non-coding RNAs such as microRNAs.

With the rapid development of high-throughput m^6^A sequencing technology in recent years, accumulated evidence has supported that m^6^A and its related factors are involved in the hematopoietic development of bone marrow by regulating the self-renewal, proliferation and differentiation of pluripotent stem cells in the bone marrow hematopoietic microenvironment. Identically, emerging studies [4,7,15,16] have indicated that abnormal m^6^A modification is closely associated with the development and progression of hematologic malignancies. In this paper, we systematically review the progress of research on the biological characteristics of m^6^A methylation modifications, their effects on normal hematopoietic regulatory functions, and their role in hematologic malignancies. It is expected to provide a scientific basis for the development of novel molecularly targeted therapies based on the aberrant m^6^A modifications in related hematologic tumors.

## 2. Biological Features of m^6^A Methylation Modifications

### 2.1. Overview of m^6^A Methylation

RNA epigenetic modifications are an important part of RNA regulation [17]. Currently, more than 150 RNA modifications have been identified in eukaryotes, mainly occurring on mRNA, tRNA, rRNA, and other non-coding RNAs, which are regulated at the post-transcriptional level [18]. Of note, the m^6^A is a methylation modification located on the sixth nitrogen atom of RNA adenine, which is the most common type of modification in higher eukaryotic mRNAs [19,20], and the number of m^6^A modified adenines accounts for 0.2 to 0.5% of the total number of adenines [21]. Moreover, m^6^A modification is widely found not only in mammals but also in yeast, plants, protozoa, and various viruses [22,23]. With the rapid development of high-throughput sequencing technology, it has been recently found that m^6^A is sequentially distributed in the protein-coding sequence (CDS), near the stop codon and in the 3’ untranslated region (3’UTR) [24,25,26], i.e., mainly localized in the highly conserved sequence of mRNA RRACH motif (R = G/A; H = U/A/C) [27,28,29]. In addition, m^6^A methylation modifications are widely distributed in mRNA and non-coding RNA (ncRNA) and play important roles in the metabolism of many RNAs, including mRNA splicing, processing and maturation of microRNA (miRNA), and long non-coding RNA (lncRNA)-mediated transcriptional repression [30,31,32].

Precisely, m^6^A was first discovered in the mRNA of hepatocellular carcinoma cells in 1974 [20,33] and has since been detected in a variety of eukaryotic organisms such as mice and yeast whose biological functions involve cell differentiation, mitosis, immune homeostasis, and other aspects. As a dynamic and reversible epistatic modification, m^6^A possesses many recognition proteins, which can be classified into three major categories: “Writers”, “Erasers”, and “Readers”, according to their functions [11] (Figure 2). Importantly, the enzymatic reaction process is divided into three modification states in sequence according to the classification: (1) mRNA is methylated by the writer (m^6^A methyltransferase); (2) the process can be reversed by the eraser (m^6^A demethylase); and (3) the methylated mRNA is recognized by the reader (m^6^A binding protein) [32]. To sum up, the specific mechanism of m^6^A is illustrated as follows.

### 2.2. m^6^A-Related Enzymes

#### 2.2.1. m^6^A Methyltransferase (Writers)

The m^6^A methyltransferase is also known as the writer [34]. The core components of the m^6^A methyltransferase complex, which performs m^6^A methylation, include three main species: METTL3 (methyltransferase-like 3), METTL14 (methyltransferase-like 14), and WTAP (Wilms’ tumor 1-associating protein) [35,36,37]. In fact, the m^6^A methyltransferase complex adds m^6^A methylation to the target mRNA through the methyl group on S-adenosylmethionine (SAM) transferase [38,39]. Among them, METTL3, with its N-terminal structural domain, methyltransferase structural domain, and two tandem zinc finger structural domains, is the main catalytic structure; on the other hand, METTL14 has a strong interaction with METTL3, but its methyl donor site is degraded and does not have catalytic activity, so it only plays a catalytic auxiliary role for METTL3 and maintains METTL3 activity [30,40]. METTL3–METTL14 heterodimer formation can induce m^6^A deposition on mammalian nuclear RNA [41,42]. Moreover, WTAP is not catalytically active, but it can help localize the complex to the nucleus and promote affinity between the complex and mRNA [43,44,45].

Indeed, METTL3 can catalyze the methylation modification of most mRNAs in vivo, and its homologue, METTL16 (methyltransferase-like protein 16), regulates the m^6^A modification of U6snRNA and a small fraction of mRNAs [46,47]. In addition, there are METTL5 (methyltransferase-like 5), ZCCHC4 (zinc finger CCHC-type containing 4), etc., which also have some catalytic functions for m^6^A on rRNAs [48]. In the methyltransferase complex, some new components including VIRMA (vir-Like m^6^A methyltransferase associated) [49,50,51], RBM15/15B(RNA binding motif protein 15/15B) [52], ZC3H13 (zinc finger CCCH-type containing 13) [53,54], etc. can interact with WTAP directly or indirectly as regulatory subunits, which together with WTAP form the MACOM complex and direct the localization of the whole complex in the cell [55,56]. Similarly, the activity of the methyltransferase complex is also susceptible to the influence of miRNAs [57].

Currently, it is generally known that m^6^A is co-transcriptionally modified on transcripts, and then METTL3 is recruited to chromatin in a transcription-dependent manner, mediating the methylation of nascent transcripts, and that the degree of methylation is strongly correlated with the activity of RNA polymerase II [58]. However, how the methyltransferase complex recognizes the substrate RNA and how the activity of the complex is regulated remain to be further investigated.

#### 2.2.2. m^6^A Demethylases (Erasers)

The m^6^A demethylases are called erasers, and their role is to catalyze the removal of methyl from m^6^A that has been methylated. Mechanistically, m^6^A methyltransferases and demethylases affect post-transcriptional gene expression levels by regulating dynamic changes in m^6^A methylation modifications of mRNA sequences. In fact, the two types of m^6^A demethylases identified so far are FTO (fat mass and obesity-associated gene) [59] and ALKBH5 (alkB homolog 5) [60]. It has been reported that FTO is an obesity-associated protein with the role of catalyzing m^6^A demethylation [59,61]. FTO belongs to the α ketoglutaric acid (α-KG) and divalent iron ion Fe(II) dependent ALKB dioxygenase family, which is localized in both the nucleus and cytoplasm of cells [61]. It possesses two structural domains: the N-terminal domain, which is conserved with the dioxygenase family sequence, and the C-terminal domain, which mainly plays a scaffolding role. The mechanism by which FTO affects the demethylation of m^6^A is that m^6^A is first oxidized to hm^6^A (N^6^-hydroxymethyladenosine) and f^6^A (N^6^-formyladenosine), then formaldehyde and formic acid are removed to complete the demethylation. Ultimately, adenine is produced [40,62].

However, the substrate of FTO is not only m^6^A but also N^6^,2’-*O*-dimethyladenine (m^6^Am) and m^1^A on tRNA are its catalytic substrates [11]. For example, in 2019, Guifang Jia’s group obtained the crystal structures of the human FTO nucleic acid complexes and resolved the interaction mechanism of FTO with each substrate [63]. It was found that although the FTO active pocket can recognize molecules of multiple RNA modification substrates, it prefers to bind N^6^-methyladenine bases and has the same demethylation activity for internal m^6^A and m^6^Am with the same RNA sequence (N^6^-methyladenine bases), indicating that the demethylation activity of FTO is mainly dependent on the recognition of residues and bases in its catalytic pocket rather than on the ribose ring and that different RNA sequences and tertiary structures also affect the catalytic function of FTO. Thus, these findings provide an understanding of the catalytic mechanism of how FTO demethylates a variety of substrates and provide potential research directions for future selective chemistries for cancer therapy. 

In recent studies, ALKBH5 was identified to be localized in the nucleus, and either silencing or overexpression of ALKBH5 could alter m^6^A levels [60]. More importantly, ALKBH5, similar to FTO, is also a member of the divalent iron ion, α-KG-dependent ALKB dioxygenase family. However, unlike FTO, the demethylation regarding the m^6^A modification site of target genes mediated by ALKBH5 does not require a multi-step reaction but is performed directly. At present, it has been indicated that ALKBH5 may regulate the out-of-nucleus transport of mRNAs [11], which is ultimately involved in the development of multiple diseases [64]. Therefore, as an m^6^A demethylase, it has been conclusively demonstrated that the knockdown of ALKBH5 in mouse models does not affect health status other than causing impaired spermatogenesis in mice, which makes ALKBH5 a potential therapeutic target in the future [65,66]. Collectively, the discovery of methyltransferases and demethylases confirms that m^6^A modification is dynamically reversible and that both work together to maintain a dynamic balance of cellular m^6^A methylation and demethylation.

#### 2.2.3. m^6^A Binding Proteins (Readers)

Currently, the m^6^A-binding protein is recognized as the reader. Specifically, m^6^A-binding proteins can regulate the biological behavior and function of mRNA by reading m^6^A methylation, and the YTH domain family, which binds to RNA, is the first of the “Readers” found to directly recognize m^6^A [67]. YTH family proteins, as reading proteins of m^6^A, are mainly divided into two isoforms, i.e., a total of five family members: YTH domain family protein (YTHDF1/2/3) and the nucleus member YTH domain-containing (YTHDC1/2) in humans. At present, the mechanism of m^6^A recognition by the YTH domain is relatively well understood [67,68]. The YTH domain contains several conserved aromatic amino acids, and the benzene ring on it not only interacts with m^6^A adenine and methyl groups but also forms a cage-like tertiary structure, which firmly holds the methyl group of m^6^A, thus recognizing m^6^A [69]. In terms of functional studies, YTH domain family proteins play a rather important role in regulating RNA stability [67], translation efficiency [68], etc. For instance, YTHDF1/2/3 are dominantly found in the cytoplasm, and the binding of YTHDF1 and YHTDF3 to each other can promote the translation efficiency of m^6^A-modified mRNAs to facilitate protein synthesis and eventually mediate the mRNA degradation due to YTHDF2 [11]. Meanwhile, YTHDF2 mediates the degradation of target gene mRNA to modulate the stability of mRNA, thereby regulating the level of gene expression in somatic cells. Notably, it can directly recruit and bind to the SH domain of the scaffold subunit CNOT1 of the CCR4-NOT (the carbon catabolite repression 4-negative on TATA-less complex) deadenylase complex and subsequently transfer to the RNA containing the m^6^A modification to degrade that RNA, or it can use the heat-responsive protein HRSP12 (heat-responsive protein 12) as a bridge to shear the interior of the RNA by RNaseP/MRP (ribonuclease P and RNase MRP) nucleases [55,70].

In 2019, Gao et al. [71] revealed that a segment of the disordered region upstream of the YTH domain in the YTHDF protein, i.e., the low complexity domain (LCD), is capable of m^6^A-dependent phase separation, which may be related to the stress phenomenon [71]. However, recently Zaccara et al. [72] proposed that YTHDF1/2/3 are not involved in the regulation of translation, and their protein structures are nearly identical, suggesting a non-specific recognition of m^6^A modifications for RNA degradation in a redundant manner. In addition, an analysis conducted by another laboratory using the QTLs(quantitative trait loci) significantly showed that m^6^A has a complex effect on translation and does not simply promote or repress it [73].

Interestingly, YTHDC1, belonging to the YTH family, is a protein localized to the nucleus. It can interact with the precursor mRNA splicing factors serine/arginine-rich splicing factor 3 (SRSF3) and SRSF10 [74] and then binds to the m^6^A site of mRNA to regulate the selective splicing of mRNA in the nucleus. Furthermore, the mRNA exit factor NXF1 binds closely to SRSF3 and promotes the exit of m^6^A-modified mRNAs from the nucleus [74,75]. On the other hand, Li et al. [76] found that YTHDC2 is present in the cytoplasm, and it not only promotes the translation of m^6^A-modified mRNA but also accelerates the degradation of m^6^A-modified mRNA. In conclusion, YTHDC2 is firmly believed to regulate the translation efficiency and stability of RNA.

Apart from the above, there are other factors that are also involved in the mechanism of m^6^A modification, and more still need to be investigated in the future. Among them, one class of binding proteins belongs to the family of heterogeneous nuclear ribonucleoproteins (HNRNPs), including HNRNPC/G and HNRNPA2/B1. In particular, HNRNPA2/B1 possesses two tandem RNA recognition motifs (RRM) and a low complexity region (LC) at the C-terminus [77]. As an intranuclear m^6^A reader, HNRNPA2/B1 recognizes the RGm^6^AC motif (the m^6^A core motif RGAC) and regulates the selective shearing of RNA in a METTL3-dependent manner. Moreover, it is also able to mediate and facilitate the processing of precursor miRNAs; when knockdown of HNRNPA2/B1 occurs, the variable shearing of cells will be abnormal, ultimately leading to a significant reduction of mature miRNAs [78]. Similarly, the RGG motif (Arg-Gly-Gly) in the LC region of HNRNPG can directly bind to the phosphorylated carboxy-terminal domain (CTD) of RNA polymerase II and recognize m^6^A modifications near the nascent RNA splice site, finally interacting with the nascent RNA to regulate the selective splicing of transcripts [79]. Overall, m^6^A modifications can reshape RNA structure and subsequently regulate RNA–protein interactions around or near it, which is referred to as the “m^6^A-switch” [80].

In recent years, it has been shown that insulin-like growth factor 2 messenger RNA (mRNA)-binding proteins (IGF2BPs) are identified as an additional class of readers. As the conserved family of single-stranded RNA-binding proteins (RBPs), it structurally contains six RNA-binding domains, including two RRM domains and four K-homology (KH) domains [81]. It has been reported experimentally that the KH3-4 domains of IGF2BPs proteins are essential for the recognition of m^6^A-modified mRNAs in vitro. In addition, IGF2BPs have the function of stabilizing mRNA, which may be achieved by their ability to recruit some RNA stabilizing proteins such as ELAV-like RNA-binding protein 1(ELAVL1), matrin3(MATR3) to function [82]. Furthermore, some recent studies have also explored other binding proteins, e.g., fragile X mental retardation protein (FMRP) [83], proline-rich coiled-coil 2A (PRRC2A) [84], eukaryotic initiation factor 3 (eIF3) [85], etc., which have a role in regulating RNA stability and RNA exit from the nucleus. However, the specific mechanisms need to be further studied.

### 2.3. m^6^A Modifications in Virus-Host Interaction

It has recently been revealed that abnormalities in m^6^A modification play an essential role in the development of human diseases, especially in diseases caused by pathogenic viruses. For instance, some viruses that pose a serious threat to human health, including human immunodeficiency virus type 1 (HIV-1), Epstein–Barr virus (EBV), hepatitis B virus (HBV), hepatitis C virus (HCV), human t-cell leukemia virus (HTLV), and human herpes virus 8 (HHV 8), act as a major cause of disease [86,87,88,89]. Upon their infection of human host cells, their RNAs are all subjected to varying degrees of m^6^A modifications. Thus, m^6^A modifications are involved in modulating the life cycle of viruses by affecting the process of their replication. Mechanistically, m^6^A modification in host cells affects the replication of pathogenic viruses and the expression of related important genes; simultaneously, the immune response of the host to viruses is also greatly influenced by m^6^A methylation. Importantly, some viral infections, such as HIV-1, EBV, HTLV, etc., can contribute to the occurrence of some hematologic malignancies (e.g., lymphoma, leukemia, etc.) [89].

## 3. m^6^A Methylation Modification and Normal Hematopoietic Regulation

To date, further studies on the function of m^6^A-related enzymes confirmed that abnormal activity of these genes led to abnormal expression of thousands of genes, firmly suggesting an important role of m^6^A in RNA metabolism. Specifically, m^6^A may affect lipid metabolism [90], sperm development [91], tumorigenesis, stem cell-directed differentiation [92], cellular reprogramming [93], biological clock rhythms, cell division, memory, and neurodevelopment [94], as well as several other life processes. Therefore, given the critical role of m^6^A in the regulation of epigenetic events, these classes of enzymes have been identified to be involved in a variety of cellular life activities, particularly in the normal regulation of hematopoiesis in the hematological system (Figure 3).

### 3.1. m^6^A Methylation Modification Regulates Hematopoietic Stem/Progenitor Cell Differentiation

Indeed, the hematopoietic system guarantees a lifelong source of blood cells, which are often derived from rarely occurring multipotent hematopoietic stem/progenitor cells (HSPCs) populations with long-term self-renewal capacity and differentiation potential [95]. The mature blood cells that differentiate from HSPCs are of two main types: myeloid and lymphoid. Within these two categories, the former include granulocytes, erythroid cells, monocytes, and megakaryocytes, while the latter contains T lymphocytes, B lymphocytes, and natural killer cells (NK cells). The hematopoietic process of the body normally begins in early embryonic development and is strictly regulated during this time. Genetic alterations such as genetic mutations and chromosomal translocations during the hematopoietic process can lead to the development of hematologic tumors if the differentiation of HSPCs is blocked or their proliferation is abnormal [96].

Recent studies have demonstrated that RNA m^6^A methylation is closely related to normal hematopoiesis and related processes in vertebrates [97]. Vertebrate HSPCs are generated from hemogenic endothelial (HE) cells through the endothelial-to-hematopoietic transition (EHT) process [98]. In 2017, Zhang et al. [99] found in zebrafish blood and vascular tissues that m^6^A could regulate the differentiation process of HSPCs by affecting the balance of HE cells gene expression during EHT through YTHDF2-mediated degradation of notch receptor 1α (Notch1α) mRNA. Furthermore, emerging evidence also indicated that deletion of *METTL3* in vascular endothelial cells significantly represses the EHT process by reducing the level of m^6^A methylation modification of Notch1αmRNA, thereby hindering the generation of HSPCs. Hence, m^6^A modulates the production of HSPCs during the hematopoietic process identified early in zebrafish embryogenesis. Additionally, a similar phenotype was observed in mouse models with knockdown of *METTL3* [45,100]. All of the above findings indicate that m^6^A methylation modifications are tightly associated with the homeostatic regulation of gene expression in HE cells.

More recently, another study [101] showed that the role of METTL3 is remarkable for the development of mammalian hematopoiesis. In the *Vav*-Cre^+^-*Mettl3^fl/fl^* (*vcMettl3^−/−^*) mouse model, deficiency of METTL3 and m^6^A resulted in hematopoietic failure accompanied by the proliferation of Lin^−^Sca-1^+^c-Kit^+^ (LSK) HSPCs in the fetal liver that is defective in hematopoiesis. Based on further analysis, deletion of m^6^A can lead to significantly upregulated transcription of the interferon-stimulated genes (ISGs) and 2′,5′-oligoadenylate synthetase (*Oas*) genes, which induces a poor double-stranded RNA (dsRNA)-mediated innate immune response. Conversely, by utilizing m^6^A modification, i.e., METTL3 supplementation, the OAS-RNase L and PKR-eIF2α signaling pathways, and the upregulation of the dsRNA sensor MDA5 and RIG-I in HSPCs could be suppressed, thereby preventing the formation of endogenous dsRNA and the deleterious innate immune response, ultimately avoiding hematopoietic failure and perinatal lethality. Similar to METTL3, METTL14 [102] is a key component of the m^6^A methyltransferase complex and is significantly increased in expression in HSPCs. However, silencing *METTL14* significantly promotes cell myeloid differentiation during the development of normal CD34^+^ HSPCs cells. Taken together, this result highlights the critical role of METTL14 and m^6^A modifications in normal hematopoiesis.

Of course, the m^6^A eraser (e.g., FTO) is dispensable for normal HSPCs. At present, it has been reported that overexpression of *FTO* [103] in R-2HG-sensitive leukemic cells but not normal CD34^+^ HSPCs attenuates R-2-hydroxyglutarate-(R-2HG)-induced glycolysis inhibition, thereby promoting leukemogenesis in vivo. In addition, as an m^6^A demethylase, knockdown of *ALKBH5* has been proven not to affect normal hematopoiesis and the function of HSPCs in the steady-state of the organism [65,66]. In contrast, compared to normal HSPCs or other types of stem cells, Shen et al. [65] revealed that ALKBH5 deficiency significantly inhibits tumor proliferation in many types of tumors, such as breast cancer, brain tumors, and acute myeloid leukemia (AML). In fact, the role of the m^6^A mRNA reader YTHDF2 in hematopoiesis is highly complex. Furthermore, accumulating evidence [104,105,106] has shown that YTHDF2 deficiency facilitates the massive expansion of HSPCs in human cord blood and mice and demonstrates for the first time the function of YTHDF2 in the maintenance of adult hematopoietic stem cells (HSCs), thus providing therapeutic potential for future clinical applications. Interestingly, based on the analysis of a recent study on young mice, Mapperley et al. [107] also demonstrated that YTHDF2 deficiency causes the failure of HSCs during serial transplantation and the prolonged activation of pro-inflammatory pathways which in turn eventually leads to progressive hematopoietic loss. Thus, this study emphasizes the importance of YTHDF2 in the long-term maintenance of HSCs expansion. Apart from YTHDC2, another study [108] identified that YTHDC1 is also essential for maintaining the normal hematopoiesis and functional development of HSPCs in vivo. In an experiment on three mouse models, i.e., *Ythdc1^fl/+^* (WT), *Ythdc1^fl/+^*Mx1Cre (*Ythdc1* HET), and *Ythdc1^fl/fl^*Mx1Cre (*Ythdc1* KO), whole blood cells as well as mature cells (e.g., myeloid, B cells, and T cells) in the serum of *Ythdc1* KO mice were markedly reduced, and all mice died within three weeks. These results indicate that YTHDC1 is required for HSPC survival and that YTHDC1 deficiency leads to rapid hematopoietic failure. Further studies revealed that *YTHDC1* KO but not *YTHDC1* haploinsufficiency has a significant negative effect on the maintenance of HSPCs. However, when YTHDC1 is overexpressed, it hinders the differentiation of HSPCs and increases the proliferation/self-renewal of HSPCs, which supports the oncogenic role of YTHDC1 in HSPCs. However, the molecular mechanisms of other m^6^A readers with regard to the regulation of normal hematopoiesis are still unclear.

To conclude, the above studies unveil that RNA m^6^A methylation is essential in modulating the differentiation of HSPCs.

### 3.2. m^6^A Methylation Modification Regulates the Differentiation of Bone Marrow Mesenchymal Stem Cells

Bone marrow mesenchymal stem cells (BMSCs), which are characterized by their self-renewal and multidirectional differentiation potential, are a different type of stem cells from HSPCs that exist in the bone marrow and can differentiate into a variety of bone marrow stromal cells, such as osteoblasts, chondrocytes, adipocytes, etc., under different induction conditions [109]. Currently, emerging studies have indicated that RNA m^6^A also plays a crucial role in regulating the differentiation of BMSCs. For example, Yao et al. [110] found that *METTL3* regulates the expression of Janus kinase 1 (JAK1) in BMSCs in an m^6^A methylation-dependent manner, while *YTHDF2* affects the expression level of JAK1 by regulating the stability of *JAK1* gene mRNA. Meanwhile, in this study, it has been proposed that reducing the m^6^A level of *JAK1* gene mRNA in BMSCs can be achieved by regulating the expression of Janus kinase 1/signal transducer and activator of transcription 5/the promoter of CCAAT/enhancer binding proteinβ (JAK1/STAT5/C/EBPβ) signaling pathway to promote the differentiation of BMSCs to adipocytes. In another study, Cen et al. [111] revealed that TRAF4 could negatively regulate adipogenesis in mesenchymal stem cells (MSCs). Recently, it has been reported that TRAF4 binds to PKM2 and activates β-catenin signaling by activating the kinase activity of PKM2, which then inhibits adipogenesis in vitro and in vivo. Additionally, the expression of *TRAF4* is reduced during adipogenesis, regulated by ALKBH5-mediated RNA m^6^A demethylation.

Specifically, BMSCs belong to the multipotential stromal cells in the hematopoietic microenvironment, which are involved in regulating the homeostasis of the bone marrow hematopoietic microenvironment. Through the synthesis and secretion of various hematopoietic factors, they can contribute to the self-renewal, proliferation, and differentiation of HSPCs, thus maintaining the balance of the hematopoietic process. In addition, osteoblasts not only are capable of maintaining the activity of HSPCs but also participate in maintaining the normal hematopoietic function of the body. In the *METTL3* knockout mouse model constructed by Wu et al. [112], mouse skeletons showed a pathological phenotype associated with osteoporosis, suggesting that *METTL3*-mediated m^6^A methylation modifications play a critical role in regulating the differentiation of BMSCs. Mechanistically, the above study also observed that parathyroid hormone/parathyroid hormone receptor-1 (PTH/PTH1R) is a downstream signaling pathway of m^6^A, and the deletion of *METTL3* inhibits PTH1R expression only at the translational level, which in turn indicates that m^6^A can affect osteogenic and lipogenic differentiation of BMSCs by positively regulating the PTH/PTH1R signaling pathway.

### 3.3. m^6^A Methylation Modification Regulates Cellular Reprogramming

Cellular reprogramming, on the one hand, refers to the process by which terminally differentiated cells are reversed under specific conditions and then restored to a totipotent state, ultimately either forming an embryonic stem cell line or developing further into a new individual. On the other hand, under embryonic stem cell (ESC) culture conditions, induced pluripotent stem cells (iPSCs) are defined as: mouse embryonic fibroblast (MEF) cells that are transferred with a total of four transcription factors [e.g., octamer-binding transcription factor 4 (Oct4), SYR-box 2 (Sox2), v-myc avian myelocytomatosis viral oncogene homolog (c-Myc), and Kruppel-like factor 4 (Klf4)] were subsequently induced into cells with ESC characteristics [113]. Actually, Chen et al. [57] showed that in MEF cells that express four pluripotency factors *Oct4, Sox2, c-Myc*, and *Klf4*, overexpression of *METTL3* significantly increases m^6^A levels; moreover, the expression of *OCT4, SOX2, and NANOG* homologous frame protein pluripotency factors was also increased in *METTL3* overexpressing MEF cells, indicating that m^6^A may modulate the expression levels of pluripotency factors in somatic cells, thereby promoting the reprogramming of somatic cells into pluripotent stem cells.

Simply put, c-MYC is a major regulator of self-renewal and differentiation in normal hematopoietic and leukemic cells. In particular, another recent finding [114] has confirmed that *METTL3* can identically enhance the expression of c-MYC protein in leukemic cells by means of increasing the level of m^6^A modification of *c-Myc* gene mRNA, thereby inhibiting cell differentiation and promoting the self-renewal of leukemic cells, and exerting oncogenic effects in hematological tumors. Conversely, the deletion of METTL3 could increase the level of phosphorylated AKT, which contributes to promoting differentiation and apoptosis in METTL3-deficient cells. Overall, these results provide a research rationale for METTL3 exploration regarding the regulation of HSPCs function or the treatment of myeloid leukemia [114].

## 4. m^6^A Methylation Modifications and Hematological Malignancies

Hematologic malignancies, a heterogeneous group of malignancies, occur as a result of malignant transformation of HSPCs, which lose the ability to further differentiate and mature, and thus are blocked at different nodes [115]. They can be divided into two types: myeloid and lymphocytic. Furthermore, the specific disease classification consists of the following four major areas: various types of leukemia, multiple myeloma, malignant lymphoma, and myelodysplastic syndromes. Among them, multiple myeloma accounts for ~10% of the overall incidence in hematologic malignancies [116]. However, the incidence of MM has steadily increased in recent years.

Recent studies have shown that RNA m^6^A methylation is closely correlated with various malignant diseases and their related processes. Here, we concentrate only on RNA m^6^A methylation in human cancers related to the hematological system. At present, a large number of studies have confirmed the considerable place of aberrant RNA m^6^A expression in the development, progression, and recurrence of different types of hematologic malignancies (Table 1).

### 4.1. The Role of m^6^A Methylation Modifications in Acute Leukemia

#### 4.1.1. RNA m^6^A Methylation and Acute Lymphoblastic Leukemia

Acute lymphoblastic leukemia (ALL) is a group of hematologic neoplasms characterized by the malignant proliferation of primitive naïve lymphocytes. It has been shown that ALL has a peak incidence in childhood (1 to 9 years) and constitutes >70% of childhood leukemias. In particular, *BCR-ABL* fusion gene-positive ALL (*BCR-ABL*^+^ ALL) accounts for 20–30% of ALL in adults and approximately 3–5% of ALL in children, with the poor effect of conventional chemotherapy, highly susceptible to relapse, and extremely poor prognosis [139].

A recent study found a high association between the expression level of m^6^A catalytic enzymes genes and disease burden in ALL. Mechanistically, using real-time quantitative polymerase chain reaction (qPCR) experiments prior to disease induction therapy, Wang et al. [117] detected increased gene expression levels of m^6^A-modified methylases (e.g., *METTL3, METTL14, WTAP*) and demethylases (e.g., *FTO and ALKBH5*) in *ETV6/RUNX1*-positive ALL patients, but decreased significantly after induction therapy. Thus, the present study suggests that dysregulation of m^6^A may also be involved in ALL. However, the complete mechanism of whether m^6^A modification levels affect the development of ALL still deserves to be further investigated. Similarly, in 2021, another recent study [119] has demonstrated that ubiquitin-specific proteases (USPs) are highly correlated with the development of t-cell acute lymphoblastic leukemia (T-ALL) and are chronically resistant to chemotherapeutic drugs. Importantly, it has been shown that ALKBH5 plays an oncogenic role in the development of human cancers, which in turn would increase the stability of USPs mRNA. In an in vitro experiment, Gong et al. [119] reported that high expression of ubiquitin-specific protease 1 (*USP1*) was associated with poor prognosis in glucocorticoid (GC)-resistant T-ALL patients. On the contrary, the knockdown of *USP1* improved CEM-C1 cells’ response to dexamethasone (Dex) sensitivity, thus contributing to promoting apoptosis and exerting oncogenic effects in ALL. Simultaneously, it was also indicated that drug resistance in T-ALL patients could be mediated by the interaction and deubiquitination between USP1 and Aurora B. Mechanistically, further studies confirmed that ALKBH5 enhanced USP1 expression via modulating USP1 RNA m^6^A demethylation levels and elevating mRNA stability; downregulation of *ALKBH5* could decrease USP1 and Aurora B levels to promote the sensitivity of CEM-C1 cells to Dex and inhibited the proliferation of tumor cells. However, when USP1 was overexpressed in CEM-C1 cells, the effect of ALKBH5 could be reversed. Furthermore, in the in vivo experiments in mice, using tail vein injection of *sh-USP1*, the growth of tumors in mice was apparently inhibited, thereby resulting in a significantly prolonged survival period, which is of reference value for clinical researches on tumor therapies.

Similarly, m^6^A modifications are essential in the pathogenesis of pediatric ALL (P-ALL). Using real-time qPCR (RT-qPCR), Sun et al. [118] detected that the expression levels of METTL3 and METTL14 are decreased in *ETV6/RUNX1*-positive P-ALL, indicating a possible role in the pathogenesis and progression of *E/R*-positive P-ALL, and thus laying the groundwork for specific treatments targeting patients with possible *E/R*-positive relapses. In two five-center case-control studies, m^6^A modifications were equally closely associated with increased tumor risk in P-ALL [140,141]. Using single nucleotide polymorphism (SNP) analysis, they found that METTL3 and METTL14 gene polymorphisms can elevate the risk of developing P-ALL, suggesting that their genetic polymorphisms may be potential biomarkers for P-ALL susceptibility and selection of oncologic chemotherapeutic drugs.

Taken together, the above results show that abnormalities of m^6^A modification-related enzymes represented by ALKBH5 may be a potential condition for the development, recurrence and drug resistance of ALL, which provides a novel research direction for its clinical diagnosis and treatment.

#### 4.1.2. RNA m^6^A Methylation and Acute Myeloid Leukemia

Acute myeloid leukemia (AML) belongs to a myeloid neoplasm with a highly heterogeneous malignant clonal proliferation of HSPCs and is the most common type of adult acute leukemia [142]. Importantly, leukemic cells are characterized by enhanced self-renewal, uncontrolled proliferation, impaired differentiation, and suppressed apoptosis, and accumulation in the bone marrow leading to a decrease in normal hematopoietic cells [143]. In spite of advances in intensive chemotherapy and allogeneic stem cell transplantation(allo-SCT), the prognosis of the majority of AML patients remains poor, with a 5-year survival rate of only approximately 24%, due to refractoriness, relapse, therapy resistance, and therapy-related mortality [144].

At the molecular level, the development of AML follows a “multiple-hit” pattern [145]. In particular, dysregulation of m^6^A-related components induces oncogene expression and thus promotes the development of malignant tumors [146]. In addition, m^6^A exerts an important effect in AML progression, as it enhances the self-renewal capacity of AML cells. A recent study presented by Sheng’s group [108] found that YTHDC1, a nuclear m^6^A reader, is highly expressed in AML and that it contributes to the maintenance of AML cell proliferation and progression; knockdown of the *YTHDC1* gene significantly blocked the proliferation of primary AML cells via modulating MCM complex-mediated DNA replication, as well as the self-renewal of LSCs in vivo in mice, thus achieving control of leukemogenesis. Collectively, this finding highlights the unique oncogenic mechanism of the m^6^A modifier YTHDC1 in AML. Consistently, another evidence [120] demonstrated that YTHDC1 is proven to be an essential m^6^A reader in myeloid leukemia by applying genome-wide CRISPR screening technology. Under the mediation of mRNA m^6^A, the production of nuclear YTHDC1-m^6^A condensates (nYACs) is markedly increased in AML cells compared to normal HSPCs, thereby maintaining the cell survival and undifferentiated state, which is essential for AML development. Additionally, it has been reported that nYACs can contribute to maintaining the stability of mRNA m^6^A and avoid degradation by PAXT complex and exosome-related RNAs. Hence, the result shows that m^6^A can maintain mRNA stability and play a vital role in the pathogenesis of AML.

In 2017, Vu et al. [114] found that METTL3 is highly expressed in AML cells and can promote AML cell proliferation, inhibit cell differentiation, and ultimately play an oncogenic role in AML through mediating m^6^A methylation modification; downregulation of *METTL3* gene can induce the differentiation and apoptosis of AML cells by increasing the phosphorylation level of phosphate protein kinase B (PKB/AKT), but not apoptosis in normal hematopoietic cells. Meanwhile, the study also found that m^6^A modification could exert oncogenic effects via elevating the translation level of related mRNAs [e.g., *c-MYC*, B-cell lymphoma/leukemia-2 (*BCL2*), phosphatase and tensin homolog deleted on chromosome ten (*PTEN*) gene, etc.] in AML cells, thus further suggesting that METTL3 can act as an oncogenic factor in myeloid malignancies, which is an important reference value for clinical treatment. Similarly, the molecular mechanism between METTL3 and adipogenesis in mesenchymal stem cells of AML patients (AML-MSCs) has been currently identified [147]. Based on the analysis of m^6^A epigenetic changes in MSCs identified by RIP-qPCR and MeRIP-qPCR, deletion of *METTL3* in AML-MSCs can facilitate increased production of AKT protein through controlling m^6^A modifications of AKT1-mRNA, leading to excessive MSC adipogenesis and consequently making AML cells resistant to chemotherapy. METTL14 is another important component of the m^6^A methyltransferase complex. Weng et al. [102] showed that METTL14 is also highly expressed in AML cells and can play a critical role in the pathogenesis of both normal myelopoiesis and AML via mediating m^6^A methylation modifications that block the differentiation of normal myeloid cells and promote malignant hematopoiesis. Overall, abnormalities in m^6^A-related factors, represented by METTL3 and METTL14, may be potential factors in the development of hematologic malignancies (e.g., AML), which provide a novel research direction for their clinical diagnosis and treatment.

Apart from the aforementioned studies in m^6^A, the role of other m^6^A modifiers [122] in the genesis of AML has been recently well-recognized. For instance, Feng et al. [122] recently provided evidence showing that YBX1, the RNA binding proteins (RBPs), is responsible for promoting the survival of myeloid leukemia cells in an m^6^A-dependent manner; conversely, its deletion dramatically induces cell death while stunting the proliferation of human and mouse AML cells in vitro and in vivo. Mechanistically, through interacting with IGF2BPs, YBX1 stabilizes m^6^A-tagged RNA. Furthermore, deletion of *YBX1* facilitates mRNA decay of *MYC* and *BCL2* in an m^6^A-dependent manner, resulting in defective survival of AML cells. Thus, the findings reveal that YBX1 is a key factor in leukemia survival, which may provide a research basis for determining whether YBX1 is a therapeutic target for AML. Moreover, other m^6^A modifiers [e.g., writers (WTAP [124,148], RBM15 [149,150]), erasers (FTO [103,128,151], ALKBH5 [65,66]), readers (YTHDF2 [105], IGF2BP1 [152]), etc.] are also crucial in influencing the occurrence and development of AML in the evidence of related studies.

### 4.2. The Role of m^6^A Methylation Modifications in Chronic Leukemia

#### RNA m^6^A Methylation and Chronic Myeloid Leukemia

Chronic myeloid leukemia (CML) is a malignant myeloproliferative neoplasm originating from multipotent hematopoietic stem cells. In essence, the t (9; 22) (q34; q11.2) translocation is a characteristic chromosomal alteration in CML, and at the molecular level, the translocation results from the formation of a BCR-ABL fusion gene [153,154]. Currently, the annual incidence of CML is (0.7–1.8) per 100,000 worldwide, which is the third-highest among all types of leukemia [155,156]. In particular, the annual incidence rate in China is approximately (3.6 to 5.5) per 1 million. Depending on the disease, CML is clinically classified into three phases: chronic phase (CP), accelerated phase (AP), and blastic phase (BP or blast crisis (BC)), and patients are generally diagnosed in the chronic phase [157,158].

At present, it has been reported that METTL3, an RNA m^6^A modifier, is also involved in the mechanism of CML development [130]. Specifically, Ianniello et al. [130] identified that the m^6^A methyltransferase complex METTL3/METTL14 is highly expressed in CML patients and is essential to maintain the proliferation of primary CML cells and CML cell lines that are sensitive or resistant to tyrosine kinase inhibitors (TKIs) (e.g., imatinib). Notably, it has been reported that aberrant translation is thought to be one of the mechanisms that mediate BCR/ABL transformation and maintain the leukemic phenotype of CML cells [159,160]. Here, the study proved that METTL3 promotes the proliferation of tumor cells by indirectly affecting MYC levels or directly methylating specific mRNAs to modulate ribosome levels and translation, resulting in tumorigenesis. Conversely, METTL3 deficiency drastically damages the translation efficiency of mRNAs for metabolism-related genes in organisms. Therefore, the findings suggest that METTL3 is a novel oncogene in the pathogenesis of CML and a potential therapeutic target for TKIs-resistant CML. Another recent finding [131] also revealed that via modulating the miR-766-5p/CDKN1A axis, METTL3 could modify long non-coding RNAs (lncRNAs) nuclear-enriched transcript 1 (NEAT1), thus controlling the progressive process of CML disorder and playing an oncogenic role in CML. NEAT1 is currently known to have low expression levels in CML cell lines or PBMCs of CML patients [131]. Mechanistically, the dysfunction of m^6^A methyltransferase METTL3 causes the abnormal expression of NEAT1 in CML, and the overexpression of NEAT1 greatly promotes the death of CML cells. Moreover, using methylated RNA immunoprecipitation (MeRIP) assay and SRAMP, we could identify the modification sites of m^6^A. Overall, this study explains the role of m^6^A in CML and further provides a rationale for targeting therapy aimed at CML. Consistently, Lai et al. [132] also observed that dysregulation of METTL3 contributes to chemoresistance of CML cells. Based on experimental analysis of mouse models and two cell lines, KCL22 and K562, the investigators confirmed that LINC00470 is able to importantly suppress the expression of PTEN mRNA through METTL3, showing a negative regulatory relationship between LINC00470 and PTEN. Conversely, the expression level of PTEN is increased after silencing METTL3 in cancer cells. Therefore, we can conclude that METTL3 is a detrimental factor in affecting the chemotherapeutic effect of CML. In this regard, we can address the corresponding molecular targets for anticancer therapy.

### 4.3. The Role of m^6^A Methylation Modifications in Lymphoma

#### 4.3.1. RNA m^6^A Methylation and Diffuse Large B-Cell Lymphoma

Diffuse large B-cell lymphoma (DLBCL) is the most prevalent subtype of non-Hodgkin’s lymphoma (NHL), accounting for approximately 30–35% of NHL. DLBCL is a heterogeneous tumor that can develop at any age, with a peak age of 50–70 years. Along with the aging of the population, the incidence of lymphoma in the elderly is increasing year by year, and >50% of DLBCL patients are >60 years old [161].

Importantly, m^6^A modifiers also play a crucial role in the progression of DLBCL. At present, it is generally recognized that PIWI-interacting RNAs (piRNAs) exert oncogenic effects as epigenetic effectors in human cancers [135]. In a xenograft DLBCL model, depletion of piRNA-30473 can suppress the proliferation of tumor cells. Han et al. [135] further discovered that piRNA-30473 could cause the upregulation of WTAP expression and enhance the levels of DLBCL m^6^A, which in turn results in the development of DLBCL disease. In addition, the upregulation of WTAP facilitates the expression of its critical target gene, hexokinase 2 (*HK2*), and therefore contribute to the progression of DLBCL, as verified by the experimental method m^6^A sequencing. To sum up, the study emphasizes that the RNA m^6^A methylation factor, i.e., WTAP, possesses an important function in DLBCL and will support the exploration of novel therapeutic approaches for DLBCL. As an m^6^A writer, METTL3 is another potential target for the treatment of DLBCL [134]. By regulating m^6^A modifications in pigment epithelium-derived factor (PEDF), METTL3 is functionally closely associated with the development of DLBCL. Indeed, METTL3 is significantly overexpressed in DLBCL tissues and cell lines, ultimately promoting the proliferation of tumor cells. However, overexpression of PEDF reverses the inhibitory effect of DLBCL cell proliferation due to METTL3 deficiency. Hence, the above findings suggest that METTL3 is essential in boosting the progression of DLBCL. Strikingly, a recent study of Xie’s group has clearly demonstrated a correlation between m^6^A modifications and the prognosis of DLBCL [133]. Based on the expression of m^6^A modifiers, the researchers distinguish different subtypes of DLBCL at the molecular level using a consensus clustering approach. Thus, by affecting m^6^A regulators correlated with survival, they observed that different DLBCL patients will possess adverse prognostic characteristics and outcomes of disease progression. Further, based on an analysis of prognostic correlations, it was determined that in DLBCL patients, high-risk m^6^A modifications signify more rapidly progressive tumors, thereby indicating poorer patients’ survival. From this point of view, we firmly believe that m^6^A-related modifiers can be well-suited indicators to determine the prognostic status of DLBCL. However, further elucidation of the risk assessment mechanism underlying m^6^A modifiers for tumor disease is still required.

#### 4.3.2. RNA m^6^A Methylation and Other Types of Lymphoma

Apart from DLBCL mentioned above, there are currently relatively few mechanistic studies involving the relationship between m^6^A and other less frequent types of lymphoma [e.g., natural killer (NK)/T-cell lymphoma, B-cell lymphoma, T-cell lymphoblastic lymphoma (T-LBL), etc.]. For example, T-LBL belongs to a group of highly aggressive hematologic malignancies that originate from primitive T lymphocytes. To our knowledge, the pathogenesis of T-LBL was first confirmed in 2021 by An et al. [162] to involve modulation based on m^6^A modifications. In line with its role in other tumors, the m^6^A methyltransferase METTL14 is also essential in influencing the development of T-LBL. Numerous evidence shows that microRNAs (miRNAs) play a vital regulatory role in human cancers. Using high-throughput sequencing technology, they further reveal that miR-211 is a miRNA with down-regulated expression in T-LBL, and silencing miR-211 predicts poor disease outcome; however, overexpression of miR-211 remarkably reduces the viability and inhibits the proliferation of T-LBL cells. Mechanistically, METTL14 enables methylation modification of primary miR-211 (pri-miR-211), which unexpectedly exerts anti-cancer effects. Similarly, in xenograft tumor models, miR-211 significantly suppresses the growth of tumors in vivo. To conclude, these data definitively suggest that targeting *miR-211*, a target gene, may be a promising therapeutic approach for patients with T-LBL. Another study [163] has revealed that the occurrence of NK/T-cell lymphoma (NKTCL) is also importantly correlated with RNA m^6^A methylation. Compared to normal NK cells in the organism, the expression level of WTAP is markedly elevated in human NKTCL cell lines (e.g., YTS and SNK-6 cells). By contrast, WTAP deficiency notably inhibits the proliferation of YTS and SNK-6 cells and promotes the death of tumor cells. Specifically, Ma et al. [163] recently found that WTAP increases the m^6^A level of dual-specific phosphatase 6 (DUSP6) mRNA in an m^6^A-dependent manner, thereby enhancing its expression and ultimately causing the oncogenic effects of NKTCL. Simultaneously, WTAP also heightens the chemoresistance to the drug cisplatin. These results highlight that the treatment of NKTCL may possibly be achieved via modulating WTAP.

In addition, it has been reported that B-cell lymphomas are similarly involved in m^6^A methylation modifications. Wu et al. [164] showed that, either in vitro or in vivo experiments, suppression of ALKBH5 or overexpression of selected MYC-repressed genes (MRG) SPI1 or PHF12 could effectively impede the growth of tumor protein MYC-mediated B-cell lymphoma. Moreover, in this study, emerging evidence shows that MYC also modulates the translation of MRG mRNA through a particular m^6^A reader, YTHDF3. Notably, given that ALKBH5 and YTHDF3 are major representatives of m^6^A binding proteins, the findings reflect that MYC promotes the development of tumors via altering the m^6^A modification of target genes. Consistently, m^6^A reader YTHDF1 is currently identified to inhibit EBV replication in patients and play a vital anti-cancer role, thus providing a new target for future treatment of EBV-infected B-cell lymphoma [165]. To our knowledge, mantle cell lymphoma (MCL) is a highly aggressive non-Hodgkin B-cell lymphoma with a median age of diagnosis of 60 years. By analyzing the gene expression omnibus database, Zhang et al. [166] found that approximately half of the m^6^A regulators can predict the survival of patients with MCL, especially the m^6^A.index, and can classify them. Concretely, MCL patients with low m^6^A.index have lower survival rates and lower mRNA levels in the total transcriptome. In essence, it is probably due to the link of m^6^A regulators with cell division and RNA metabolic pathways. In summary, the study suggests that imbalanced expression levels of m^6^A regulators may be used to predict poor survival in patients with MCL, thus providing an idea for treatment.

### 4.4. The Role of m^6^A Methylation Modifications in Multiple Myeloma

Multiple myeloma (MM) is a malignant hematologic disease with abnormal proliferation of clonal plasma cells, which can destroy the bones, kidneys, and hematopoietic function of the body. Among them, renal injury is a common complication of MM and one of its major causes of death [167]. In a current study, a computational method, xPore, based directly on the analysis of RNA sequencing (RNA-seq), in the absence of matched control samples, showed that only one m^6^A locus manifests a remarkable difference in the rate of modification in six cell lines and MM patient samples [168]. This result demonstrates that xPore can identify differential m^6^A modifications and expression from a high-throughput experiment for the data in question.

Indeed, FTO is also a promising target for the treatment of MM [137]. Functionally, isocitrate dehydrogenase 2 (IDH2) regulates mRNA m^6^A modification by activating FTO; on the other hand, it can down-regulate the m^6^A level of WNT7B mRNA and increase the expression of WNT7B, thus activating the Wnt signaling pathway, which ultimately facilitates the growth of tumor cells in MM in vitro and shortens the overall survival of patients. It is apparent that m^6^A and its related factors (e.g., FTO) are involved in the progression of MM and provide a potential therapeutic target. Furthermore, ALKBH5 is similarly confirmed to be overexpressed in MM and predicts a worse prognosis for MM patients [136]. Functionally, the downregulation of ALKBH5 promotes tumor cell death and suppresses the growth of MM cells in vitro, exerting a tumorigenic effect. Thus, the results suggest that ALKBH5 is essential in maintaining the progression of MM tumors and provide a novel direction for scientists to investigate. Interestingly, HNRNPA2B1, a specific m^6^A reader, is also found to be essential in the pathogenesis of MM. In 2021, Jiang et al. [138] concluded that HNRNPA2B1 is highly expressed in MM patients and mediates m^6^A methylation modification of ILF3 mRNA, thereby contributing to the proliferation of MM cells in vitro and in vivo, which results in MM patients presenting a poor prognosis; conversely, genetic depletion of HNRNPA2B1 inhibits the growth of tumor cells and exerts an anti-cancer effect. Taken together, the above findings reflect that m^6^A modification may be of great importance for the occurrence of MM and provide a possible research direction for clinical researchers.

### 4.5. The Role of m^6^A Methylation Modifications in Myelodysplastic Syndromes

Myelodysplastic syndrome (MDS) is defined as a myeloid clonal disorder of hematopoietic stem cell origin with symptoms including ineffective hematopoiesis, intractable anemia and a high risk of AML. Current studies generally agree that the occurrence and development of MDS is a cumulative damage of multiple genetic events and that the combination of genomic abnormalities and altered epigenetic modifications can cause phenotypic heterogeneity of MDS [169]. Among these, altered expression levels of relevant genes may affect key oncogenes or tumor suppressor genes in MDS myelopoietic cells, which allow abnormal hematopoietic cells to gain a clonal proliferative advantage. Previous studies [170] have pointed out that genes encoding RNA spliceosomes may be involved in the development and progression of MDS as early driver genes. Importantly, small alterations in the spliceosome can affect the specificity of splicing and trigger changes in the peptide sequences of transcribed proteins, which subsequently affect the normal physiological functions of hematopoietic cells. However, the exact mechanism of MDS induced by aberrant RNA splicing is hitherto unknown.

Actually, YTHDC1 can mediate the mRNA splicing process by recruiting precursor mRNA splicing factors [74], which can contribute to elucidating the role of splicing abnormalities in the abnormal hematopoietic function of MDS. In addition, the reduced level of m^6^A modification induced by downregulation of the *METTL3* gene can modify the alternative splicing (AS) pattern of precursor mRNA, which is mainly enriched in the p53 signaling pathway and apoptosis pathway [24,26]. Therefore, the above results further illustrate that m^6^A methylation modification is important for the exploration of the abnormal splicing mechanism for MDS. The m^6^A-based regulation of bone marrow hematopoiesis plays an important role in maintaining the normal hematopoietic process of bone marrow. Bone marrow is the main site of hematopoiesis in the body, and abnormal regulation of bone marrow hematopoiesis can potentially lead to the development of a series of diseases. Inherited bone marrow failure syndromes (IBMFS) are a heterogeneous group of genetic disorders characterized by common clinical features of bone marrow hematopoietic failure, congenital malformations, and high risk of progression to malignancy [171], mainly including Fanconi anemia (FA), dyskeratosis congenita (DC), Diamond–Blackfan anemia (DBA), and Shwachman–Diamond syndrome (SDS). Among them, FA is due to defective DNA repair mechanisms, while DC is caused by telomerase defects, both of which have a tendency to acquire genetic alterations and develop into MDS. Specifically, the hematologic lesions of IBMFS manifest as impaired bone marrow hematopoiesis and decreased whole blood cells or mono-lineage cells, implying that IBMFS lesions are derived from the myeloid pluripotent stem cell level. m^6^A and its related factors could be involved in the development and progression of IBMFS via dramatically affecting the development of bone marrow hematopoiesis. Nevertheless, the exact mechanism of action remains to be elucidated through further studies.

## 5. Discussion

It is widely recognized that m^6^A modification is a widespread form of methylation modification in eukaryotic RNA. Under the joint reversible regulation of m^6^A methyltransferases, m^6^A demethylases and m^6^A binding proteins, m^6^A methylation modification covers various stages of RNA life course such as mRNA, lncRNA, and miRNA. Through such epigenetic m^6^A methylation modifications, various RNAs (e.g., mRNA, lncRNA, and miRNA) are modulated to perform their corresponding functions, which in turn alter the expression of tumor cell-related genes and thus facilitate the development of hematological tumors. Notably, studies related to the effects of m^6^A and its related factors on hematologic malignancies (e.g., AML, DLBCL, MM, etc.) are currently a hot topic in the research of RNA epigenetics. However, the researches on the species and functions of m^6^A-related proteins are still in the exploratory stage. The existing studies demonstrate that: (1) m^6^A methylation modification is a double-edged sword, either over-modification or under-modification may result in the development of tumor; (2) the same m^6^A methylation-related enzymes may possess different functions in different action pathways; (3) m^6^A methylation modification is not only limited to mRNA but also exists in lncRNA, miRNA and other RNAs, which provides a wide scope for our research; (4) m^6^A methylation-related enzymes can also act as the regulatory targets of lncRNAs and miRNAs, which affect the m^6^A methylation levels of other RNAs; (5) In addition to modulating m^6^A levels, m^6^A methyltransferases probably also possess other independent functions. For instance, METTL3, besides having methyltransferase activity, can enhance the translation of cancer genes and affect the progression of tumors independently of their catalytic subunits. These findings also extend the research horizon of m^6^A for us. 

Explicitly, remarkable progress has been made in exploring the role of m^6^A methylation in hematological malignancies. However, many opportunities and challenges exist: (1) The role of m^6^A in hematological tumors is still controversial, and these functional features are attributed to the fluctuating distribution of m^6^A in different regions of mRNA and subcellular readers. Given the extensive interactions between metabolic networks, it is particularly important to maintain the balance of the internal environment between metabolic processes, which is one of the factors to be considered in the future; (2) whether m^6^A and its regulators can be used as potential biomarkers for diagnosis and prognosis of hematologic tumors, and the specificity and sensitivity of these biomarkers remain to be explored; (3) some studies indicate that m^6^A regulators and related pathways can be considered as therapeutic targets, but specific applications in large samples of clinical practice are deficient and their side effects are largely unknown; (4) Whether there is a combined or subtractive effect between m^6^A methylation modifications and other epigenetic modifications needs to be further explored; (5) Most of the existing hematological tumor studies have focused on methyltransferases and demethylases, but there are fewer studies on readers, which is one of our future directions to explore.

## 6. Conclusions

Collectively, based on the above findings, m^6^A methylation modification can remarkably affect the self-renewal, proliferation, and differentiation processes of pluripotent stem cells in hematologic malignancies, but their exact biological functions in normal hematopoietic and malignant hematologic diseases are not clear. Simultaneously, the molecular mechanism of m^6^A methylation modification that regulates the hematopoietic system and induces hematological tumors needs to be further elucidated. This is significant for our in-depth understanding of the pathogenic mechanism in hematological malignancies and the exploration of new targeted interventions. Therefore, we are convinced that with continuous research on the functional changes after m^6^A modification of RNA, a reference basis for clinicians to diagnose hematologic malignancies early and search for therapeutic targets will be provided.

## Figures and Tables

**Figure 1 cancers-14-00332-f001:**
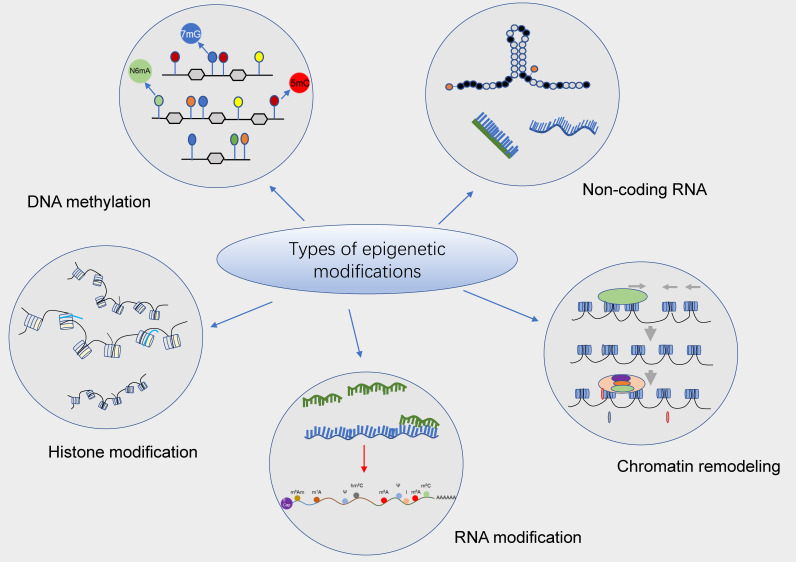
Potential epigenetic regulatory mechanisms in hematologic malignancies. Epigenetic modifications are heritable alterations that can generate changes in gene activity independent of changes in gene nucleotide sequences, including DNA methylation, histone modifications, chromatin remodeling, non-coding RNA, and RNA and RNA modifications, etc., which primarily mediate changes in gene transcription as well as translation activity. Numerous studies have confirmed that epigenetic modifications (e.g., DNA methylation and histone modifications) play an essential role in the development and progression of hematologic malignancies and are considered to be a vital target for the treatment of different types of leukemias and other hematologic malignancies.

**Figure 2 cancers-14-00332-f002:**
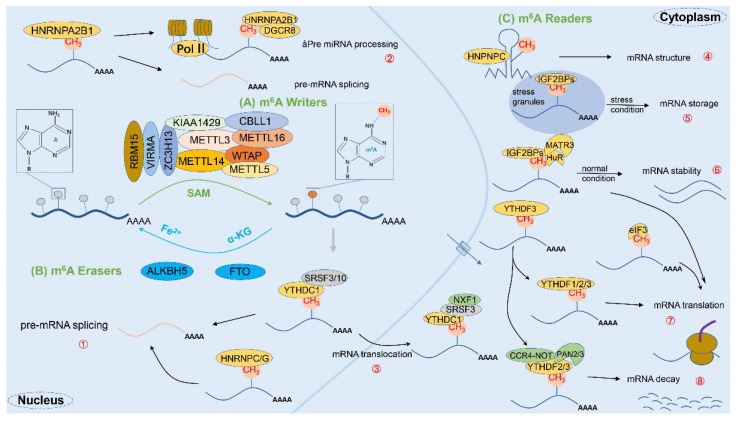
The complete process of m^6^A methylation modification. The m^6^A methylation modification is a dynamic and reversible enzymatic process and is strictly modulated by writers, erasers, and readers. (**A**) First, mRNA methylation is catalyzed by writers (e.g., m^6^A methyltransferases METTL3, METTL14, WTAP, RBM15, VIRMA, and ZC3H13, etc.); (**B**) Second, the process can be reversed by erasers (e.g., m^6^A demethylases FTO and ALKBH5); (**C**) Finally, the methylated mRNA will be recognized by readers (e.g., m^6^A binding proteins YTHDF1/2/3, YTHDC1/2, HNRNPs, IGF2BP, eIF3, etc.), thereby fulfilling the corresponding physiological roles. Specifically: ① mRNA splicing: YTHDC1, HNRNPA2/B1, HNRNPC/G, etc.; ② miRNA processing: HNRNPA2/B1; ③ mRNA translocation: YTHDC1; ④ mRNA structure: HNPNPC; ⑤ mRNA storage: IGF2BPs; ⑥ mRNA stability: IGF2BPs; ⑦ mRNA translation: YTHDF1/2/3, YTHDC2, IGF2BPs, eIF3, etc.; ⑧ mRNA decay: YTHDF2/3, YTHDC2.

**Figure 3 cancers-14-00332-f003:**
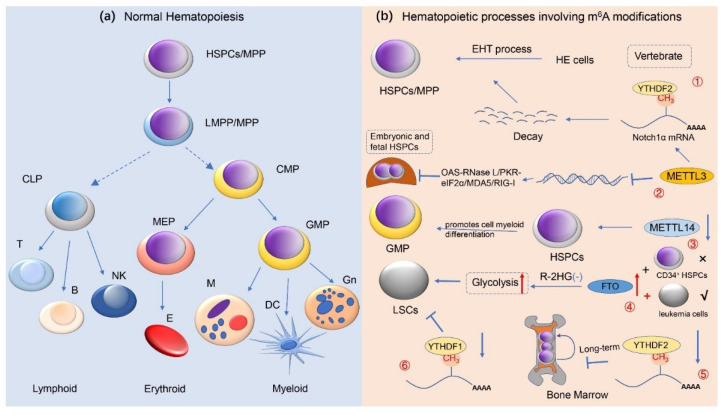
m^6^A methylation modifications and normal hematopoietic regulation. This image depicts the role of RNA m^6^A methylation in normal human hematopoiesis. (**a**) The development and differentiation process with respect to human normal hematopoietic stem/progenitor cells (HSPCs) is presented here, i.e., from HSPCs to terminally differentiated erythroid, lymphoid, and myeloid cells; (**b**) It has revealed that m^6^A modification modulates the normal hematopoietic system. ① METTL3-mediated m^6^A modifications regulate HSPC fate specification via suppressing Notch signaling during early definitive hematopoiesis. ② During the expansion of HSPC in the fetal liver, METTL3 deletion promotes the formation of dsRNA, activates the OAS RNase L and PKR-eIF2a pathways, and upregulates MDA5/RIG-I, resulting in hematopoietic failure. ③ METTL14 is highly expressed during the development of normal CD34^+^ HSPCs cells, and silencing METTL14 facilitates terminal myeloid differentiation of HSPCs cells. ④ In R-2HG-sensitive leukemic cells but not in normal CD34^+^ HSPCs, overexpression of FTO reverses the effect of R-2HG-induced glycolysis inhibition, leading to leukemogenesis in vivo. ⑤ Deficiency of YTHDF2 causes the failure of hematopoietic stem cells during serial transplantation and prolonged activation of pro-inflammatory pathways, ultimately contributing to progressive bone marrow bias. ⑥ YTHDC1 deletion impedes the proliferation and survival of LSCs in vivo, supporting the oncogenic role of YTHDC1 in leukemias (e.g., AML). Abbreviations: HSPCs hematopoietic stem/progenitor cells; MPP multipotent progenitors; LMPP lymphoid-primed multipotential progenitors; CMP common myeloid progenitor; CLP common lymphoid progenitor; GMPs granulocyte/monocyte progenitor; MEPs megakaryocytes/erythroid progenitor; Gn granulocyte; DC dendritic cell; M monocyte; E erythrocyte; NK natural killer cell; EHT endothelial-to-hematopoietic transition; LSCs leukemia stem cells; HE hemogenic endothelial; R-2HG R-2-hydroxyglutarate.

**Table 1 cancers-14-00332-t001:** The functional roles of RNA m^6^A methylation modification in various types of hematologic malignancies.

Cancer Type	m^6^A Modifiers	Patients/Cell Lines	Role of m^6^A in Cancer	Functions	Mechanism	References
ALL	Writers/Erasers	In *ETV6/RUNX1*-positive ALL patients	oncogene	High levels of m^6^A “writers” (METTL3, METTL14, WTAP) and m^6^A “erasers” (FTO and ALKBH5) mRNA expression prior to induction therapy resulted in a high disease burden in ALL patients	Not available	[117]
	METTL3/METTL14	In childhood *ETV6/RUNX1*-positive P-ALL	oncogene	The decreased levels of METTL3 and METTL14 indicate a possible role in the pathogenesis and progression of *E/R*-positive P-ALL.	Not available	[118]
	ALKBH5	In glucocorticoid (GC)-resistant T-ALL patients/CEM-C1 cells/mice	oncogene	Inhibition of ALKBH5-mediated m^6^A modification decreased *USP1* expression, and downregulation of USP1 ameliorated GC resistance in T-ALL by suppressing Aurora B expression and elevating GR levels	ALKBH5/USP1/Aurora B	[119]
AML	YTHDC1	In human AML cells/LSCs in vivo in mice	oncogene	YTHDC1 is overexpressed in AML, and it contributes to the proliferation and survival of human HSPCs/AML cells, as well as the self-renewal of leukemia stem cells (LSCs) in mice	YTHDC1/ MCM4	[108]
	YTHDC1	In AML cells	oncogene	nYACs maintain mRNA stability, as well as AML cell survival and undifferentiated state; moreover, nYACs protect m^6^A-mRNA from degradation by PAXT complex and exosome-related RNAs	YTHDC1-m^6^A condensates (nYACs)	[120]
	YTHDF2	In leukemic cells in vitro and in mice/in AML patients	oncogene	Overexpressed in t (8; 21)-type AML patients; when YTHDF2 is knocked down, it inhibits tumor cell proliferation in vitro and in mice	AML1/ETO-HIF1α loop/YTHDF2/TNFRSF1b	[121]
	YTHDF2	In mouse and human AML	oncogene	YTHDF2 contributes to the initiation of AML disease as well as proliferation and maintains the overall functional integrity of LSCs	YTHDF2/ TNFR2	[105]
	YBX1/IGF2BPs	Primary human and mouse AML cells in vitro and in vivo	oncogene	Expression of YBX1 is markedly upregulated in myeloid leukemia cells, and YBX1 deficiency greatly induces apoptosis and promotes differentiation while reducing proliferation and impairing leukemic competence of primary human and mouse AML cells in vitro and in vivo	YBX1/MYC/BCL2 (mRNA)	[122]
	METTL3	In MOLM-13 cell lines/recipient mice in vivo	oncogene	METTL3 is highly expressed in AML cells as well as promotes AML cell proliferation and inhibits cell differentiation; m^6^A modification elevates the translation levels of *c-MYC*, *BCL2,* and *PTEN* gene mRNAs in AML cells	METTL3/c-MYC/BCL2/PTEN (mRNA)	[114]
	METTL3	In AML cells and immunodeficient mice	oncogene	In AML cells, METTL3 promotes tumor cell proliferation and inhibits cell differentiation; downregulation of METTL3 results in the inability of immunodeficient mice to develop leukemia.	CEBPZ/ METTL3/ SP1	[123]
	METTL14	In normal HSPCs and AML cells	oncogene	METTL14 is overexpressed in AML cells and can block the differentiation of normal myeloid cells and promote malignant hematopoiesis via m^6^A modifications	SPI1-METTL14-MYB/MYC	[102]
	WTAP	In AML patients/WTAP knockout AML cells	oncogene	WTAP promotes AML cell proliferation, tumorigenesis, and inhibits cell differentiation. In addition, WTAP causes chemoresistance in AML cells	WTAP/MYC mRNA	[124]
	WTAP	In AML patients or in AML cells in vitro in vivo	oncogene	miR-550-1 leads to a further decrease in WWTR1 stability by downregulating the expression level of WTAP, which ultimately disrupts AML cell proliferation and tumorigenesis	miR-550-1/WTAP/ WWTR1	[125]
	WTAP	In different AML cell lines, e.g., K562 cell line	oncogene	Under the regulation of functional METTL3, the expression of WTAP is upregulated and promotes the proliferation of AML cells	METTL3/WTAP	[126]
	FTO	In vitro, in mice, primary patient cells, and TKI-resistant cells	oncogene	SsD inhibits AML cell proliferation and promotes apoptosis and cell cycle arrest via targeting FTO/m^6^A signaling both in vitro and in vivo	Not available	[127]
	FTO	In AMLs	oncogene	FTO enhances leukemia oncogene-mediated cell transformation and leukemogenesis and suppresses all-trans retinoic acid (ATRA)-induced AML cell differentiation and apoptosis	FTO/ASB2, RARA	[128]
	FTO	In (R-2HG-sensitive) leukemia cells	oncogene	R-2HG abrogated FTO/m^6^A/YTHDF2-mediated post-transcriptional upregulation of PFKP and LDHB (two key glycolytic genes) expression, thereby attenuating aerobic glycolysis in leukemia	FTO/m^6^A/PFKP/LDHB axis	[103]
	FTO	In human AML cell lines and AML patients	oncogene	FTO inhibitors, namely FB23 and FB23-2, inhibit proliferation and promote differentiation/apoptosis in human AML cells and primary cells	Not available	[129]
	ALKBH5	In human AML LSCs	oncogene	By regulating the chromatin state of the ALKBH5 locus, the expression of ALKBH5 can be elevated, thereby maintaining leukemogenesis in human AML	KDM4C, MYB, Pol II /ALKBH5/AXL Signaling Axis	[66]
	ALKBH5	In human AML/in LSCs/LICs	oncogene	ALKBH5 not only facilitates the proliferation of AML cells, but also contributes to the self-renewal of leukemic stem/initiating cells (LSCs/LICs)	ALKBH5/TACC3	[65]
CML	METTL3	In CML patients/CML cell lines	oncogene	Depletion of METTL3 strongly impairs the translation efficiency of mRNA and contributes to the proliferation of CML cells	METTL3/PES1 protein	[130]
	METTL3	PBMCs and CML cell lines	oncogene	Overexpression of NEAT1 inhibits cell viability and promotes apoptosis in CML cells	METTL3/NEAT1/miR-766-5p/CDKN1A axis	[131]
	METTL3	In a mouse model, and in KCL22 and K562 cells	oncogene	Dysregulation of METTL3 promotes chemoresistance and inhibits autophagy in CML cells	LINC00470/METTL3/PTEN mRNA	[132]
DLBCL	m^6^A regulators	In DLBCL patients	oncogene	In patients with DLBCL, high-risk m^6^A indicates worse survival when grouped according to prognostic characteristics	Not available	[133]
	METTL3	In DLBCL tissues and cell lines	oncogene	METTL3 promotes tumor cell proliferation	METTL3/ PEDF	[134]
	WTAP	In xenograft DLBCL models	oncogene	piRNA-30473 facilitates the proliferation of DLBCL cells and induces cell cycle arrest via upregulating WTAP	piRNA-30473/WTAP/HK2 m^6^A	[135]
MM	ALKBH5	in MM cells, xenograft models or patients	oncogene	ALKBH5 deficiency induces apoptosis and inhibits the growth of MM cells in vitro	ALKBH5/ TRAF1/NF-κB and MAPK	[136]
	FTO	in CD138^+^ cells from MM	oncogene	IDH2 promotes the growth of myeloma cells in vitro by targeting FTO to regulate the m^6^A RNA level of MM	IDH2/FTO/WNT7B/Wnt	[137]
	HNRNPA2B1	in MM patients and in MM cells	oncogene	Overexpression of HNRNPA2B1 promotes the proliferation of MM cells in vitro and in vivo	HNRNPA2B1/ILF3 mRNA/AKT3	[138]
MDS	YTHDC1	In MDS cells	oncogene	Causes abnormalities in hematopoietic function	YTHDC1/SRSF3 or SRSF10	[74]

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
