# Peer review of "The Role of N6-Methyladenosine (m6A) Methylation Modifications in Hematological Malignancies"

_cancers, 2022, doi:10.3390/cancers14020332_

Round 1

Reviewer 1 Report

In this manuscript, entitled "The role of N6-methyladenosine (m6A) methylation modifications in hematological malignancies- -Stage1”, authors review the mechanisms of m6A methylation and describe its role in normal hematopoietic regulation and its deregulation in several types of hematological malignancies. The field of epitranscriptomics in general, and m6A methylation in particular, is a very interesting topic in Oncology with relevant implications for several types of tumors. However, some parts of the manuscript should be improved.

Specific comments:

  • What do the authors mean with “- -Stage1” in the title?
  • The following paragraph is relevant because authors explained the concept of epigenetics, but this concept needs to be more clearly explained, avoiding redundant information: “…epigenetics has been identified as a novel concept that corresponds to genetics and encompasses many types of well-documented epigenetic modifications. In plain words,epigenetic modification is a form of gene expression regulation that affects gene transcription and translation without changes in nucleotide sequence, and can regulate and thus affect gene expression at the level of DNA and chromatin structural modifications, RNA stability, and transcriptional activity, including DNA methylation modifications, histone covalent modifications, chromatin remodeling, non-coding RNA regulation, RNA modifications, etc.[9-11].” The concept of “epitranscriptomics” or “epitranscriptome” should be introduced and explained in the manuscript. This term is related to m6A and should be used for example in line 83 instead of “epigenome”.
  • In line 86 and 176, “Similar to DNA methylation” should be avoided, since RNA modification is a different type of epigenetic modification.
  • Figure 1 is unclear. Authors are describing epigenetic modifications, therefore, gene mutations should be removed from the figure. In addition, the explanation related to genetic alterations should be removed from the figure legend. The picture that represents the DNA methylation is not clear. What do the different colors indicate? Importantly, the figure that represents the RNA modifications is unclear: the m6A methylation in RNA is not represented in the figure.
  • Figure 2: In general, this figure should be improved to understand better the molecular mechanisms related to m6 The figure legend should be also explained in more detailed to be able to follow well the figure. Otherwise, the term “(Figure 2)” should be indicated in the text of the manuscript when a concept include in the figure is mentioned. Most relevant abbreviations should be included in figure legend.
  • What does “RGm6AC” mean? The term “RGm6AC” should be defined the first time it appears in the text.
  • Figure legend of Figure 3 should be explained in more detailed. Most relevant abbreviations should be included in figure legend.
  • Line 288-296; line 435-441: some reference is needed.
  • Some expressions should be revised: “No study” (in Table 1), “From the other hand” (line 320)…The term “In the present study” (line 571) it is not properly used.
  • Table 1: the same format should be used in all the columns. For example, in column “m6A modifiers”, it is possible to see uppercase or lowercase letters at the beginning of the sentence. At least, the most relevant abbreviations should be included at the bottom of the table.

Author Response

Response to  Reviewer 1 Comments:

Point 1: What do the authors mean with “- -Stage1” in the title?

Response 1: Thank you very much for your comments. The term "stage 1" indicates the first stage of a manuscript submission, which has currently been deleted.

Point 2: The following paragraph is relevant because authors explained the concept of epigenetics, but this concept needs to be more clearly explained, avoiding redundant information: “…epigenetics has been identified as a novel concept that corresponds to genetics and encompasses many types of well-documented epigenetic modifications. In plain words,epigenetic modification is a form of gene expression regulation that affects gene transcription and translation without changes in nucleotide sequence, and can regulate and thus affect gene expression at the level of DNA and chromatin structural modifications, RNA stability, and transcriptional activity, including DNA methylation modifications, histone covalent modifications, chromatin remodeling, non-coding RNA regulation, RNA modifications, etc.[9-11].” The concept of “epitranscriptomics” or “epitranscriptome” should be introduced and explained in the manuscript. This term is related to m6A and should be used for example in line 83 instead of “epigenome”.

Response 2: Thank you very much for your suggestion. We are very sorry for the incomplete discussion in “epitranscriptomics”. The concept has been introduced in the manuscript. (Line 71-76) In addition, we have changed the “epigenome” to “epitranscriptomics”. (Line 88)

Point 3: In line 86 and 176, “Similar to DNA methylation” should be avoided, since RNA modification is a different type of epigenetic modification.

Response 3: The section on “Similar to DNA methylation” has been revised. (Line 91, 183)

Point 4: Figure 1 is unclear. Authors are describing epigenetic modifications, therefore, gene mutations should be removed from the figure. In addition, the explanation related to genetic alterations should be removed from the figure legend. The picture that represents the DNA methylation is not clear. What do the different colors indicate? Importantly, the figure that represents the RNA modifications is unclear: the m6A methylation in RNA is not represented in the figure.

Response 4: Thank you very much for your comments. Figure 1 has been re-corrected and the section on “gene mutations” has been removed from the Figure 1. (1) DNA methylation is a commonly observed epigenetic phenomenon, which refers to the addition of methyl groups to bases (e.g., A/G/C) in DNA molecules via the action of DNA methyltransferases (DNMTS). In Figure 1, different colors indicate different forms of DNA methylation, such as 5-methylcytosine (5mC), N6-methyladenine (N6mA) and 7-methylguanine (7mG). (2) The content of RNA modifications has also been corrected.

Point 5: Figure 2: In general, this figure should be improved to understand better the molecular mechanisms related to m6A. The figure legend should be also explained in more detailed to be able to follow well the figure. Otherwise, the term “(Figure 2)” should be indicated in the text of the manuscript when a concept include in the figure is mentioned. Most relevant abbreviations should be included in figure legend.

Response 5: Thank you very much for your comments. According to your suggestion, this figure has been improved, and the figure legend has also been reinterpreted in the manuscript.

Point 6: What does “RGm6AC” mean? The term “RGm6AC” should be defined the first time it appears in the text.

Response 6: Thank you very much for your suggestion, and we have done it according to your ideas. RGm6AC: the m6A core motif RGAC. The m6A mark is known to occur within the mRNA RGAC motif, with R representing a purine. Meanwhile, it has been reported that HNRNPA2B1 also recognizes this motif. (In line 272) (PMID: 592376; PMID: 26321680) Thanks again!

Point 7: Figure legend of Figure 3 should be explained in more detailed. Most relevant abbreviations should be included in figure legend.

Response 7: Thank you for your comments. Figure legend of Figure 3 has been explained in more detailed, and the relevant abbreviations have been added in the manuscript. (Line 324-342)

Point 8: Line 288-296; line 435-441: some reference is needed.

Response 8: Some references have been re-added in the two sections. (Lines 313-322; Lines 497-504)

Point 9: Some expressions should be revised: “No study” (in Table 1), “From the other hand” (line 320) … The term “In the present study” (line 571) it is not properly used.

Response 9: Currently, these expressions have been revised in the manuscript. We have changed the " No study " to " Not available "(in Table 1), “From the other hand” to “Furthermore” (line 362), and “In the present study” to “At present” (line 646).

Point 10: Table 1: the same format should be used in all the columns. For example, in column “m6A modifiers”, it is possible to see uppercase or lowercase letters at the beginning of the sentence. At least, the most relevant abbreviations should be included at the bottom of the table.

Response 10: The format in all the columns has been corrected in Table 1. Additionally., the most relevant abbreviations (e.g., m6A modification-related enzymes) have been stated in the previous content. 

Reviewer 2 Report

Comment

 In this issue, the authors review the role of N6 Methyladenosine methylation modification in haematological malignancies.

The present manuscript is of significant interest and relevant to therapeutic research to the field of blood malignancies . The manuscript is well written and comprehensive. This review should be accepted for publication after very minor reviews.

Manuscript status accepted

Minor correction,

Figure 2

Typos pri-mRNA processing--àPre mRNA processing

Line 329-334

These few lines have to be rewritten to simplify and clarify message. Make more sentences but shortened them. To avoid confusion in the message.

LINE 358-363

The authors mentioned YTHDC1 is essential for maintaining normal haematopoiesis and functional development of HSPCs in vivo. However, YTHDC1 deficiency do not impact self-renewal capacity of HSPCs. How is YTDHC1 essential for normal hematopoiesis then?  What is the critical role of YTHDC1 in hematopoiesis maintenance? If YTHDC1 is essential, then something should be altered when YTHDC1 is ko/loss. Can the authors clarify?

Line 401-422

This paragraph should be deleted it is out of the topic of this review: Role of N6 Methyladenosine methylation modification in haematological malignancies.

Line 433 add reference 

Line 435-441 add references

Line 442-444 : delete this

From: In addition…to…...[109-116]

Line 470: In this study we know

Change this to : it has been showed or something more appropriate

Line 576-581

Talking about the dual effect of METTL3 deficiency on CML sensitivity to chemotherapy but also how it impacts normal haematopoiesis and promote tumorigenesis. Conclusion should be tempered, this dual effect should be cautiously mentioned. Indeed. This might be the limit to the use of METTL3 targeting drugs in clinic.

Author Response

Response to Reviewer 2 Comments:

Point 1: Figure 2 Typos pri-mRNA processing--àPre mRNA processing

Response 1: Thank you very much for your suggestion. According to your suggestion, we have changed the pri-mRNA processing to àPre miRNA processing.

Point 2: Line 329-334

These few lines have to be rewritten to simplify and clarify message. Make more sentences but shortened them. To avoid confusion in the message.

Response 2: Thank you very much for your kind reminder! In lines 371-382, this section has been re-written.

Point 3: LINE 358-363

The authors mentioned YTHDC1 is essential for maintaining normal haematopoiesis and functional development of HSPCs in vivo. However, YTHDC1 deficiency do not impact self-renewal capacity of HSPCs. How is YTDHC1 essential for normal hematopoiesis then?  What is the critical role of YTHDC1 in hematopoiesis maintenance? If YTHDC1 is essential, then something should be altered when YTHDC1 is ko/loss. Can the authors clarify?

Response 3: Thank you very much for your comments. We are very sorry for the unclear discussion about the content of this section. Currently, the content has been partially revised. In fact, YTHDC1 deficiency affects the self-renewal ability of HSPCs. This study revealed that YTDHC1 KO but not YTDHC1 haploinsufficiency has a significant negative effect on the maintenance of HSPCs. Thanks again!

Point 4: Line 401-422

This paragraph should be deleted it is out of the topic of this review: Role of N6 Methyladenosine methylation modification in haematological malignancies.

Response 4: Thank you very much for your suggestion. Here, we are very sorry for the unclear discussion about the content of this paragraph. We have re-checked the entire paragraph carefully and corrected some mistakes. The narrative of this paragraph is to illustrate the role of c-Myc gene in promoting the reprogramming of somatic cells into pluripotent stem cells, thus further explaining the function of inhibiting cell differentiation and promoting self-renewal of leukemic cells in hematologic malignancies. (Line 472-476, 482-485) Thanks again!

Point 5: Line 433 add reference

Response 5: The reference has been re-added. (Line 495, reference 114)

Point 6: Line 435-441 add references

Response 6: The reference has been re-added. (Line 497-504; references 115, 116)

Point 7: Line 442-444: delete this

From: In addition…to…... [109-116]

Response 7: Thank you for your comments. This section has been removed and modified again. (Line 505-509)

Point 8: Line 470: In this study we know

Change this to: it has been showed or something more appropriate

Response 8: The section on "In this study we know" has been revised. (Line 535)

Point 9: Line 576-581

Talking about the dual effect of METTL3 deficiency on CML sensitivity to chemotherapy but also how it impacts normal haematopoiesis and promote tumorigenesis. Conclusion should be tempered, this dual effect should be cautiously mentioned. Indeed. This might be the limit to the use of METTL3 targeting drugs in clinic.

Response 9: Thank you very much for all your suggestions. After your kind reminder, the discussion of this section has been revised accordingly in an objective and analytical manner. (Line 647-658) Thanks again!

Reviewer 3 Report

The Zhao and Peng’s review is comprehensive and well written paper focused on m6A-associated mechanisms and functions in several major hematological malignancies. The authors have summarized the dual role of m6A methylation as well as its prospects in these cancers. There have been only few similar articles published but this review is probably the first to systematize this topic from the diagnoses side of hematological malignancies.

Therefore, I recommend that this review to be published in Cancers, but the authors should address some of my comments.

Major.

  1. Some interesting papers on pediatric ALL [1-5] as well as Hodgkin and non-Hodgkin’s lymphoma [6-7] could be included in this review
  2. Some viral infections, such as EBV, can contribute to the development of some lymphomas. Moreover, some studies have shown that m6A modifications in virus-host interaction plays a significant role in regulating viral replication. Therefore, I think that adding to the review a short section on this issue could be valuable if the authors share my idea.

Minor

  1. Title: I can’t understand what does the term “stage 1” mean here?
  2. Figure 1: Putting the item "gene mutations" may be misleading 
  3. Lines 290-292 “ Specifically, m6A may affect lipid metabolism, sperm development, tumorigenesis, stem cell directed differentiation, cellular reprogramming, biological clock rhythms, cell division, memory, and neurodevelopment, as well as several other life processes.”. - the references should be added
  4. Line 307: “HSPCs are of two main types: myeloid and germline” – it should be rather myeloid and lymphoid
  5. Lines 440-441: Why only MM was distinguished by giving a detailed percentage of incidence? The reference for that is required.
  6. Line 454 “childhood ALL has a peak incidence between 0 and 9 years of age” – it should be 1 and 9 years of age. ALL in infants is rare.
  1. Luo A, Yang L, Li M, Cai M, Huang A, Liu X, Yang X, Yan Y, Wang X, Wu X, Huang K, Huang L, Liu S, Xu L, Liu X. Genetic Variants in METTL14 are Associated with the Risk of Acute Lymphoblastic Leukemia in Southern Chinese Children: A Five-Center Case-Control Study. Cancer Manag Res. 2021;13:9189-9200  https://doi.org/10.2147/CMAR.S335925
  2. Liu X, Huang L, Huang K, Yang L, Yang X, Luo A, Cai M, Wu X, Liu X, Yan Y, Wen J, Cai Y, Xu L and Jiang H (2021) Novel Associations Between METTL3 Gene Polymorphisms and Pediatric Acute Lymphoblastic Leukemia: A Five-Center Case-Control Study. Front. Oncol. 11:635251. doi: 10.3389/fonc.2021.635251
  3. Sun, C, Chang, L, Liu, C, Chen, X, Zhu, X. The study of METTL3 and METTL14 expressions in childhood ETV6/RUNX1-positive acute lymphoblastic leukemia. Mol Genet Genomic Med. 2019; 7:e933. https://doi.org/10.1002/mgg3.933
  4. Sumedha Saluja, Jay Singh, Ayushi Jain, Shilpi Chaudhary, Karthikeyan Pethusamy, Parthaprasad Chattopadhyay, Sameer Bakhshi, Anita Chopra, Archna Singh, Subhradip Karmakar, Jayanth Kumar Kumar Palanichamy; Delineating the Role of Interplay between m6A Machinery Genes and IGF2BP Group of RNA-Binding Proteins in B-Cell Acute Lymphoblastic Leukemia (B-ALL). Blood 2021; 138 (Supplement 1): 4472. doi: https://doi.org/10.1182/blood-2021-152097
  5. Jiang C, Trudeau SJ, Cheong TC, Guo R, Teng M, Wang LW, Wang Z, Pighi C, Gautier-Courteille C, Ma Y, Jiang S, Wang C, Zhao B, Paillard L, Doench JG, Chiarle R, Gewurz BE. CRISPR/Cas9 Screens Reveal Multiple Layers of B cell CD40 Regulation. Cell Rep. 2019 Jul 30;28(5):1307-1322.e8. doi: 10.1016/j.celrep.2019.06.079. PMID: 31365872; PMCID: PMC6684324.
  6. Zhang W, He X, Hu J, Yang P, Liu C, Wang J, An R, Zhen J, Pang M, Hu K, Ke X, Zhang X, Jing H. Dysregulation of N6-methyladenosine regulators predicts poor patient survival in mantle cell lymphoma. Oncol Lett. 2019 Oct;18(4):3682-3690. doi: 10.3892/ol.2019.10708. Epub 2019 Aug 2. PMID: 31516580; PMCID: PMC6732954.
  7. Esteve-Puig R, Climent F, Piñeyro D, et al. Epigenetic loss of m1A RNA demethylase ALKBH3 in Hodgkin lymphoma targets collagen, conferring poor clinical outcome. Blood. 2021;137(7):994-999. doi:10.1182/blood.2020005823

Author Response

Response to Reviewer 3 Comments:

Major.

Point 1: Some interesting papers on pediatric ALL [1-5] as well as Hodgkin and non-Hodgkin’s lymphoma [6-7] could be included in this review

Response 1: Thank you very much for your comments. According to your suggestion, we have a discussion of pediatric ALL as well as non-Hodgkin’s lymphoma in the manuscript [1-3, 6]. (Line 553-563, 754-763) In addition, [4] (63rd ASH Annual Meeting Abstracts) has not been found, and [5] is closely associated with DLBCL. Further, [7] is mainly involved in m1A and DNA methylation, which is not consistent with the topic of the manuscript. Thanks again!

Point 2: Some viral infections, such as EBV, can contribute to the development of some lymphomas. Moreover, some studies have shown that m6A modifications in virus-host interaction plays a significant role in regulating viral replication. Therefore, I think that adding to the review a short section on this issue could be valuable if the authors share my idea.

Response 2: Thank you for your comments. The section on " m6A modifications in virus-host interaction " has been added to the manuscript. (Line 297-311)

Minor

Point 3: Title: I can’t understand what does the term “stage 1” mean here?

Response 3: Thank you very much for your comment. The term "stage 1" indicates the first stage of a manuscript submission, which has currently been deleted.

Point 4: Figure 1: Putting the item "gene mutations" may be misleading

Response 4: The section on "gene mutations" has been deleted in Figure 1.

Point 5: Lines 290-292 “Specifically, m6A may affect lipid metabolism, sperm development, tumorigenesis, stem cell directed differentiation, cellular reprogramming, biological clock rhythms, cell division, memory, and neurodevelopment, as well as several other life processes.”. - the references should be added

Response 5: Thank you very much for your kind reminder! The references have been added. (Lines 315-318)

Point 6: Line 307: “HSPCs are of two main types: myeloid and germline” – it should be rather myeloid and lymphoid

Response 6: The section on “HSPCs are of two main types: myeloid and germline” has been revised. (Line 348)

Point 7: Lines 440-441: Why only MM was distinguished by giving a detailed percentage of incidence? The reference for that is required.

Response 7: Thank you very much for your comment. We distinguish MM by providing detailed incidence percentages mainly due to the fact that it has the lowest incidence among the three major malignancies (e.g., leukemia, lymphoma, MM) and have added the reference to the manuscript. (Line 503) Thanks again!

Point 8: Line 454 “childhood ALL has a peak incidence between 0 and 9 years of age” – it should be 1 and 9 years of age. ALL in infants is rare.

Response 8: Thank you very much for your suggestion. We have changed the “childhood ALL has a peak incidence between 0 and 9 years of age” to " ALL has a peak incidence in childhood (1 to 9 years)". (Line 519)

Round 2

Reviewer 3 Report

I have no further comments on this manuscript.